# Consumption of a Branched-Chain Amino Acid (BCAA) during Days 2–10 of Pregnancy Causes Abnormal Fetal and Placental Growth: Implications for BCAA Supplementation in Humans

**DOI:** 10.3390/ijerph17072445

**Published:** 2020-04-03

**Authors:** Chiu Yuen To, Muriel Freeman, Lon J. Van Winkle

**Affiliations:** 1Department of Surgery, Division of Neurosurgery, St Francis Hospital, Memphis, TN 38119, USA; tochiuyuen@gmail.com; 2Department of Biochemistry, Midwestern University, Downers Grove, IL 60515, USA; muriel.freeman@my.rfums.org; 3Department of Surgery, Division of Podiatric Medicine and Surgery, Carle Richland Memorial Hospital, Olney, IL 62450, USA; 4Department of Medical Humanities, Rocky Vista University 8401 S. Chambers Road, Parker, CO 80134, USA

**Keywords:** branched-chain amino acids, isoleucine, fetus, placenta, small-for-gestational-age, embryo, muscle, exercise training

## Abstract

A relatively large branched-chain amino acid (BCAA) supplement, consumed for more than 10 days, appears to be especially effective at alleviating muscle damage and soreness during intense human training. However, perturbations in amino acid and protein consumption could have unwanted transgenerational effects on male and female reproduction. This paper hypothesizes that isoleucine consumption by female mice from days 2 to 10 of pregnancy will alter fetal and placental growth later in gestation. Mice that had received 118 mM isoleucine in their drinking water delivered pups on day 19 of pregnancy that were 9% larger than normal, whereas the reverse was true for pups born on day 20. Moreover, the inverse correlation between birth weight and litter size was lost in mice that previously consumed excess isoleucine. Similarly, the normal correlations between fetal and placental weights were lost by day 18 of pregnancy in mice that had consumed excess isoleucine. Mice that consumed excess isoleucine had placentas smaller than, and fetuses larger than normal on day 18 of pregnancy, but the reverse was true on day 15. Other unintended and unexpected effects of BCAA consumption should be studied more thoroughly due to the increasing use of BCAAs to alleviate muscle damage and soreness in athletes.

## 1. Introduction

Dietary branched-chain amino acid (BCAA) supplementation is becoming a solution to several human health problems [1,2,3,4,5,6,7,8]. Excess consumption of BCAAs reduces muscle soreness from exercise [1,2,3], counters fatigue during exercise [4], and alleviates exercise-induced skeletal muscle damage [1,3,5]. Higher BCAA levels are also associated with a lower prevalence of obesity [6,7], and their increased consumption improves liver function in patients undergoing liver surgery [8].

Alterations in protein and amino acid intake can, however, adversely influence embryo development with transgenerational consequences in adulthood [9,10,11]. For example, the consumption of a low protein diet by pregnant rats led to the development of larger than normal fetuses on day 19 of pregnancy, but their growth was not sustained, and they were smaller than the control fetuses by day 21 of pregnancy [12]. Both small and large-for-gestational-age offspring of mammals, including humans, are predisposed to develop metabolic syndrome and related disorders in adulthood [10,11,13]. Hence, we studied whether the altered consumption of a BCAA during pregnancy can influence the growth and development of mouse embryos.

A relatively large BCAA supplement, consumed for more than 10 days, appears to be especially effective at alleviating muscle damage and soreness during intense human training [3]. In some studies, male and female athletes more than doubled their intake of BCAAs [1,2]. Other studies included an additional amino acid, such as arginine, in their supplements (e.g., [14]). For these reasons, we simplified our approach by limiting this study to a single BCAA, and we more than doubled the isoleucine (Ile) intake by mice between days 2 and 10 of pregnancy. Ile was selected, in part, because it is less biologically active than leucine, which serves as a signaling molecule via mammalian target of rapamycin (mTOR) [15]. While some amino acids are known to regulate embryo development through one-carbon and other aspects of metabolism in stem cells [11], nothing is known about how Ile might influence these and other cells.

We performed two studies: In study 1, we determined whether the consumption of Ile from days 2 to 10 of pregnancy alters the birth weights of mouse pups. Study 2 was designed to determine whether Ile consumption leads to abnormal fetal and placental growth. We hypothesized that the consumption of Ile by female mice from days 2 to 10 of pregnancy will change the fetal and placental growth later in gestation, resulting in small- or large-for-gestational-age offspring.

## 2. Methods

Sexually mature Swiss ICR female mice (Harlan Sprague-Dawley, Inc., Indianapolis, IN, USA) were allowed to acclimate to a 14 h light–10 h dark cycle for at least 2 weeks [16,17]. They were then placed with a fertile male, and natural ovulation and mating were confirmed by the presence of a copulatory plug the following morning (day one of pregnancy). Two groups of mice were included in study 1; one group of 12 experimental (E) mice received 118 mM Ile in their drinking water from days 2 to 10 of pregnancy, while another 12 served as control (C) mice and drank regular water. All mice consumed Purina rodent chow ad libitum. Pregnancies were otherwise allowed to proceed normally, and the pups delivered on days 19 and 20 were weighed immediately.

Subsequent experiments in study 2 involved four groups of 8 to 12 mice each. The pups of two groups were delivered via caesarian sections on day 15 of pregnancy, with one group (the experimental mice) having drunk 118 mM Ile-treated water from days 2 to 10 of pregnancy, and the other (the control mice) having consumed regular water. Similarly, the conceptuses from the other groups of control (C) and experimental (E) mice were obtained on day 18. Daily water and food intake were measured in the groups of mice that underwent caesarian sections on day 18. At the time of delivery, the conceptuses were carefully dissected in an attempt to preserve the integrity of the amniotic membranes for the measurement of the weights of the whole conceptuses. The weights of the whole conceptuses, placentas, and fetuses were measured upon delivery. Figure 1 displays a summary of our scheme to collect the fetal and placental weights on day 18 of pregnancy. A similar approach was used to collect data on day 15.

Data were analyzed statistically using contingency tables, t-tests, determination of Pearson correlation coefficients (*r* values), and analyses of variance (ANOVA) combined with multiple comparison tests as appropriate (GraphPad Prism 8.0.2 Software, Inc., La Jolla, CA, USA). Effect sizes were also calculated as *r* values.

Data were analyzed on both per conceptus/offspring and per dam bases, as there is controversy regarding whether the unit of dietary treatment of pregnant mice, or each of their conceptuses/offspring, is the dam [18,19,20]. In the case of per dam analysis, the mean weights of the fetuses, placentas, and offspring from a given dam were calculated, and these means were used in statistical analyses as single pieces of data. Consequently, sample sizes in the latter cases equal the number of dams, rather than the number of conceptuses/offspring. When the sample size is made larger by comparing the means for individual conceptuses/offspring, rather than the number of dams, the level of statistical significance is, of course, higher. Data were reported as means + 95% confidence intervals (CI).

These studies were approved by the Midwestern University Institutional Animal Care and Use Committee (MWU File Numbers 1486 and 1560).

## 3. Results

The experimental mice gained 28.73 + 3.49 g between days 1 and 18 of pregnancy, while the control mice gained 28.59 + 4.64 g (see food and water consumption below). In study 1, four control mice delivered pups on day 19 of pregnancy, and eight delivered pups on day 20. Conversely, eight experimental mice delivered pups on day 19 of pregnancy, while four delivered pups on day 20. The pups of the experimental (E) mice were about 9% heavier than those pups born to the control (C) mice on day 19 (E19 vs. C19 in Table 1). On day 20, however, pups born to the experimental mice were 9% lighter than pups of the control mice (E20 vs. C20 in Table 1). Thus, pups of the experimental mice were large-for-gestational-age on day 19, but they were small-for-gestational-age when born on day 20. Furthermore, birth weights were inversely correlated with litter size in the control mice (*r* = −0.65, *p* < 0.05) but not in the experimental mice (*r* = −0.05, ns).

In study 2, the fetal weights were positively correlated with the placental weights in both the control (*r* = 0.43, *p* < 0.001) and experimental (*r* = 0.48, *p* < 0.001) mice on day 15, but this correlation was lost in the experimental mice (but not the control mice) by day 18 (*r* = 0.06, ns in the experimental mice vs. *r* = 0.34, *p* < 0.001 in the control mice). The change in the *r* values for the experimental mice between days 15 and 18 was also statistically significant (Fisher r-to-z transformation, *p* < 0.001). The sizes of placentas increased in the control mice between days 15 and 18 of pregnancy, but such was not the case for the experimental mice (Figure 2). Moreover, fetuses were larger in the control mice than in the experimental mice on day 15, but the reverse was true on day 18 (Figure 3). Similarly, fetal/placental weight ratios per dam were lower in the experimental mice than in the control mice on day 15, but the opposite was true on day 18 (Figure 4).

The experimental mice also had an increased fragility of fetal membranes on day 18 of pregnancy, as indicated by the percentage of membrane ruptures during dissection. Dissections were performed carefully in an attempt to preserve the integrity of the whole conceptuses for weighing, and the investigators had no preconceived notion that such an event would occur more frequently in one group than another. When performing caesarian sections, there was a 60% greater incidence of unwanted rupturing of amniotic membranes in the experimental mice than in the control mice on day 18 (Figure 5).

Ile supplementation between days 2 and 10 of pregnancy did not greatly alter water and food intake by the mice between days 2 and 18 of gestation, as shown in Figure 6 and Figure 7. Nor did differences in water and food intake seem to account for differences in the fetal and placental weights in the control and experimental mice on day 15 of pregnancy. The increase in food intake by the experimental mice on day 17 as displayed in Figure 7 may, however, have supported more rapid than normal fetal growth between days 15 and 18 of pregnancy, shown in Figure 3.

## 4. Discussion

Ile consumption by female mice for 10 days after mating caused multiple changes later in pregnancy and even after gestation. These changes included increases and decreases in the weights of the resultant fetuses, placentas, and offspring, shown in Table 1 and Figure 2, Figure 3 and Figure 4. For pup weights, the effect sizes for ANOVA on a per dam or per offspring basis were *r* = 0.77 and 0.58, respectively, and are of crucial practical importance [21]. For the direct comparison of the control and experimental offspring born on day 20, these numerically adjacent mean values had effect size values on a per dam and per offspring basis of *r* = 0.63 and 0.50, respectively, and are also of crucial practical importance [21]. Extraembryonic membranes also appeared to become more fragile as a result of prior Ile supplementation, shown in Figure 5. Hence, the unintended effects of BCAA supplementation to support strength training in humans may occur after the period during which more BCAAs are consumed.

The current results using a mouse model may seem, at first, to apply more to reproductive-age females than male athletes. However, perturbations in amino acid and protein consumption by males also adversely affect the offspring they sire [11,22]. Moreover, the full effects of BCAA supplementation on athletes remain to be established. Although positive effects on training have been observed, studies are needed to determine whether BCAAs have immediate, as well as longer-term, detrimental impacts.

Such effects seem especially likely to occur in processes involving stem cells. Perturbations in stem cell function occur as a result of challenges to protein and amino acid metabolism and signaling [9,10,11]. The present results show that Ile supplementation may produce such challenges, as evidenced by the abnormal growth of mouse fetuses due to prior Ile consumption by their mothers.

However, by what mechanism might Ile supplementation given to mice during the first half of pregnancy alter fetal and placental growth closer to the conclusion of gestation? One possibility is the partial Ile inhibition of leucine-stimulated mTOR signaling during the preimplantation blastocyst development period [15]. Subsequently altered peri-implantation development, due to this challenge to amino acid metabolism and signaling, could lead to abnormal placental function and fetal growth later on, as is the case for low protein diets [9,10,11,15]. Such abnormal placental function likely includes inhibition of placental insulin, mTOR, and signal transducer and activator of transcription (STAT) signaling, and the resultant down-regulation of amino acid transporter expression [23,24]. The effects of excess Ile consumption are likely to be more complex; however, the experimental mouse fetuses in our study exhibited both slower and more rapid growth than the normal mouse fetuses depending on the period of development, as shown in Figure 3.

Moreover, the effects of protein and amino acid challenges are not always intuitively obvious or easy to predict. For example, maternal consumption of a low protein diet does not alter the concentration of BCAAs in rat fetuses, but the addition of threonine to the low protein diet significantly lowers the concentrations of these amino acids in the fetuses [25]. The spectra of possible effects of protein and amino acid perturbations, such as from BCAA supplementation, warrant further exploration, especially since their effects may be transgenerational [10,11].

## 5. Limitations

We studied the effects of a single BCAA on growth and development in mice. Thus, it is a challenge to extrapolate our findings to other species, or to other BCAAs and mixtures of them. Nevertheless, the developmental origins of health and disease (Barker) hypothesis applies well to all mammalian species including humans [9,10,11]. According to this well-documented theory, maternal and paternal lifestyle changes, such as an increase or decrease in dietary amino acid consumption, regulate early embryo development through both genetic and epigenetic modifications. These modifications can last a lifetime, and can be passed to future generations. Moreover, environmental challenges act through epigenetic changes in stem cells in both human and rodent embryos [9,10,11]. Hence, it seems prudent to study the effects of dietary BCAAs more broadly, in both rodent models as well as humans who consume BCAAs to improve their own health. Even the beneficial changes associated with BCAA consumption may result, in part, from altered stem cell function in adults.

## 6. Conclusions

We verified our hypothesis that Ile consumption by female mice from day 2 to 10 of pregnancy alters fetal and placental growth later in gestation. Ile supplementation led to slower than normal growth of fetuses up to day 15 of pregnancy, but then faster growth between days 15 and 18 of gestation. Conversely, Ile consumption produced large-for-gestational-age offspring on day 19 of pregnancy, but pups born on day 20 were smaller than normal. Abnormal placental development likely contributed to this atypical fetal growth pattern. We suggest that Ile supplementation, around the time of embryo implantation on day 5 of gestation, started aberrant placentation by altering leucine-signaling via mTOR.

## Figures and Tables

**Figure 1 ijerph-17-02445-f001:**
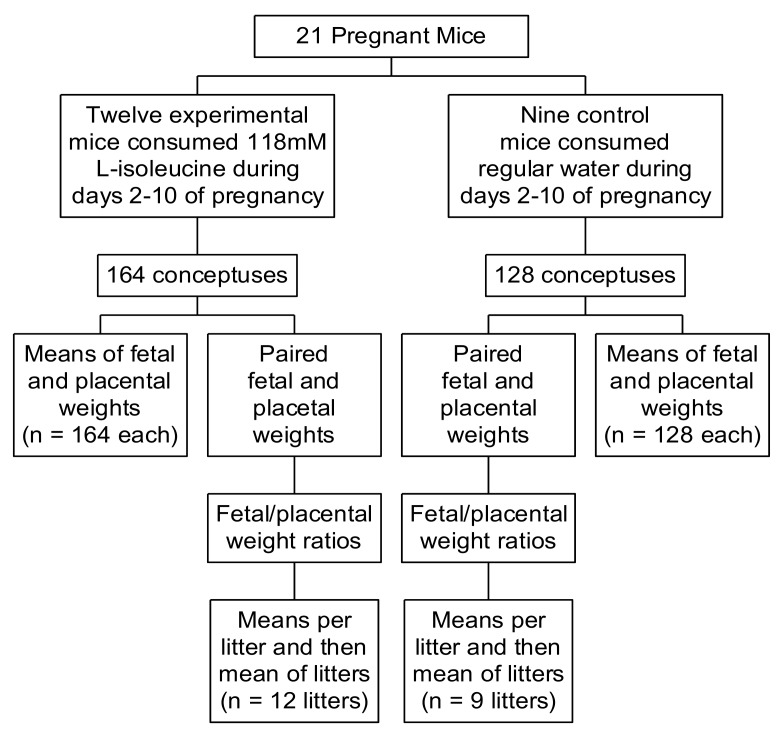
The experimental (E) and control (C) mice that underwent caesarian sections on day 18 of pregnancy.

**Figure 2 ijerph-17-02445-f002:**
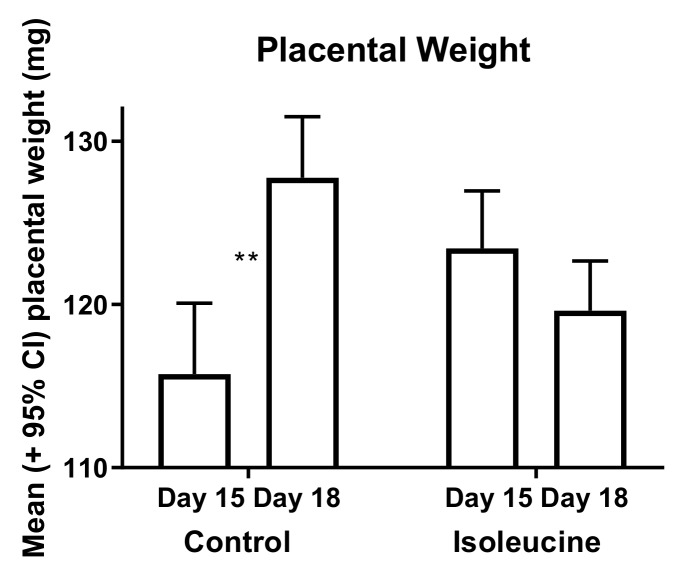
Placentas grew significantly between days 15 and 18 of pregnancy in the control mice (t-test, *p* < 0.0001), but not in the experimental mice. Double asterisks (**) indicate mean values that are significantly different from each other (control day 15, *n* = 95 placentas and day 18, *n* = 128 placentas; experimental day 15, *n* = 123 placentas and day 18, *n* = 164 placentas). CI—confidence interval.

**Figure 3 ijerph-17-02445-f003:**
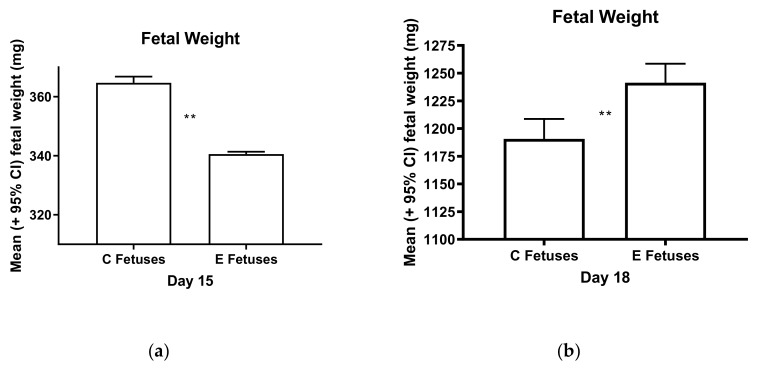
The experimental (E) fetuses were significantly smaller than the normal, control (C) fetuses on day 15 of pregnancy (**a**; t-test, *p* < 0.0001), but they were larger than normal on day 18 (**b**; t-test, *p* < 0.0001). Double asterisks (**) indicate mean values that are significantly different from each other. Also shown are the distributions of weights on day 18 for the control (C, **c**) and experimental (E, **d**) fetuses. (See Figure 2 for sample sizes).

**Figure 4 ijerph-17-02445-f004:**
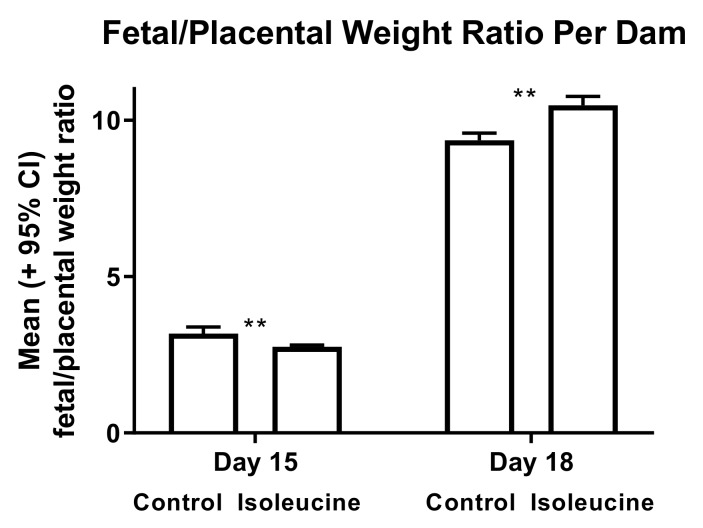
The fetal/placental weight ratio on a per dam basis was larger in the control than in the experimental (isoleucine) mice on day 15 (t-test, *p* < 0.001), but this ratio was larger in the experimental than in the control mice on day 18 (t-test, *p* < 0.0001). Double asterisks (**) indicate mean values that are significantly different from each other (control day 15, *n* = 8 dams and day 18, *n* = 9 dams; isoleucine day 15, *n* = 10 dams and day 18, *n* = 12 dams).

**Figure 5 ijerph-17-02445-f005:**
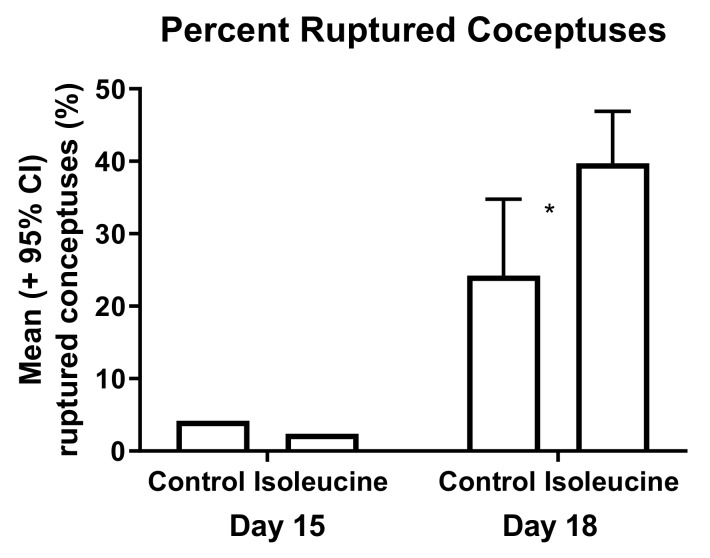
Conceptuses ruptured more frequently than normal (control) during the dissection of the experimental (isoleucine) mice on day 18 of pregnancy (contingency table, *p* < 0.01; mean ± 95% CI ruptured conceptuses per dam in control vs. isoleucine mice, t-test after arcsine transformation of the data, *p* < 0.05). A single asterisk (*) indicates mean values that are significantly different from each other. (See Figure 2 and Figure 4 for sample sizes).

**Figure 6 ijerph-17-02445-f006:**
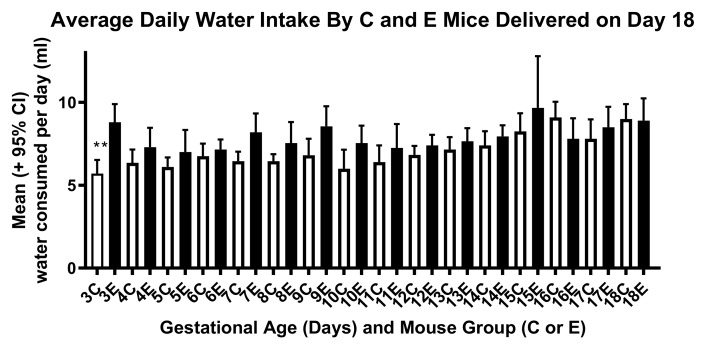
Daily water intake of the control (C) and experimental (E) mice that delivered on day 18 of pregnancy. The difference between the two groups is statistically significant only on day 3 (t-test, *p* < 0.001). Double asterisks (**) indicate mean values that are significantly different from each other (control mice, *n* = 10 dams; experimental mice, *n* = 10 dams). The number of mice is higher in Figure 4 for experimental mice because water intake was measured reliably in only 10 mice, and the number is lower for control mice in Figure 4 because one mouse delivered on day 18, before dissection could occur.

**Figure 7 ijerph-17-02445-f007:**
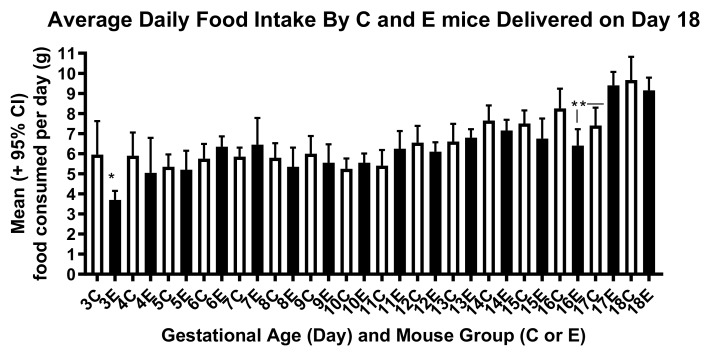
Daily food intake of the control (C) and experimental (E) mice that delivered on day 18 of pregnancy. The difference between the two groups is statistically significant on day 3 (t-test, *p* < 0.05). In addition, a significant increase in food intake occurred in the experimental mice on day 17 (ANOVA with multiple comparison tests for all E data including 16E vs. 17E, *p* < 0.0001), but such was not the case in the control mice until day 18. Single and double asterisks (*, **) indicate mean values that are significantly different from each other (control mice, *n* = 10 dams; experimental mice, *n* = 10 dams). The number of mice is higher in Figure 4 for experimental mice because food intake was measured reliably in only 10 mice, and the number is lower for control mice in Figure 4 because one mouse delivered on day 18, before dissection could occur.

**Table 1 ijerph-17-02445-t001:** Mean ± 95% confidence interval birth weights for the control (C) and experimental (E) offspring born on day 19 (C19 and E19) and 20 (C20 and E20).

	Per Offspring (g/pup)	Per Dam (g/pup/dam)
C19	1.34 ± 0.03 ^a^ *n* = 60	1.35 ± 0.10 ^a^ *n* = 4
C20	1.66 ± 0.03 ^b^ *n* =102	1.66 ± 0.06 ^b^ *n* = 8
E19	1.47 ± 0.03 ^c^ *n* =106	1.47 ± 0.08 ^c^ *n* = 8
E20	1.51 ± 0.05 ^c^ *n* = 58	1.52 ± 0.10 ^c^ *n* = 4

^a,b,c^ Superscripts indicate mean weights that are significantly different (analyses of variance (ANOVA) with multiple comparison tests, *p* < 0.0001 per offspring, *p* < 0.05 per dam).

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
