# Peer review of "Consumption of a Branched-Chain Amino Acid (BCAA) during Days 2–10 of Pregnancy Causes Abnormal Fetal and Placental Growth: Implications for BCAA Supplementation in Humans"

_ijerph, 2020, doi:10.3390/ijerph17072445_

Round 1

Reviewer 1 Report

The authors incorporated all the suggestions raised by reviewers.  

Author Response

Reviewer 1

The authors incorporated all the suggestions raised by reviewers. 

Reviewer 1 made no further suggestions.

Reviewer 2 Report

Thank you very much for allowing me to review the original article "Consumption of a branched-chain amino acid (BCAA) during days 2-10 of pregnancy causes abnormal fetal and placental growth: implications for BCAA supplementation in humans" (ijerph-765661).

The study is based on dietary branched-chain amino acid (BCAA) supplementation that is becoming a solution to several human health problems and whose health effects are not fully known.

The aim of this study is studied whether altered consumption of a BCAA during pregnancy can influence the growth and development of mouse embryos.

Introduction: This must be completed. The basis for establishing causality remains to be explained.
The hypothesis should be rewritten more clearly.
The objective should be at the end of this clearly defined section. This should also be reviewed in the summary.

Methods:
Please indicate the approval number of the study by the Ethical Committee for Animal Experimentation (Midwestern University Institutional Animal Care and Use
Committee).
Has a sample size calculation been performed?
They are 24 mice’s, 12 experimental and 12 controls. (Line 58).
However, in Figure 1, there are 21 pregnant mice. Not all mice got pregnant?. Please clarify this difference.

It should be noted that the 95% Confidence Intervals have been calculated.

Results
Experimental and control mice consumed food and gained weight at the same rates during pregnancy (data not shown). Please indicate these results.
This phrase:
"The effect sizes for ANOVA of these data on a per
dam or per offspring basis were r = 0.77 and 0.58, respectively, which are of crucial practical importance [21]. "
It is not correct in results, since it is a subjective assessment, more appropriate to the discussion. Not applicable in results.
Table 1, please if you use three decimal places in all or if you use two decimal places, use the same criteria in all the results.
At the bottom of the table, indicate which test you have used.

In Figure 2, 3 4, 5, 6 and 7 the mean plus minus 95% CI is not represented. Please check it out. Indicate at the foot of the figure the tests used.

Discussion

Please adjust to the results obtained. They should review the knowledge on this topic with the literature. As well as indicating the strengths and limitations of this study. Incorporate guidance from future research based on the results obtained.

Also include how this study can help to know the effects in humans, given the species barriers.

Please also add separate conclusion section

Author Response

Reviewer 2

Thank you very much for allowing me to review the original article "Consumption of a branched-chain amino acid (BCAA) during days 2-10 of pregnancy causes abnormal fetal and placental growth: implications for BCAA supplementation in humans" (ijerph-765661).

Thank you for your comments and suggestions.

The study is based on dietary branched-chain amino acid (BCAA) supplementation that is becoming a solution to several human health problems and whose health effects are not fully known.

The aim of this study is studied whether altered consumption of a BCAA during pregnancy can influence the growth and development of mouse embryos.

Introduction: This must be completed. The basis for establishing causality remains to be explained.

We added a sentence to address this issue near the end of the Introduction (lines 52-54).

The hypothesis should be rewritten more clearly.

The objective should be at the end of this clearly defined section. This should also be reviewed in the summary.

Our hypothesis/objective is now stated clearly at the end of the Introduction (lines 55-56) and in the abstract (lines 17-19).

Methods:

Please indicate the approval number of the study by the Ethical Committee for Animal Experimentation (Midwestern University Institutional Animal Care and Use Committee).

These numbers are provided in line 94.

Has a sample size calculation been performed?

We estimated that an average of at least six dams would be needed per group, but did not perform a formal calculation at the time of the study.  Nevertheless, a formal calculation using Mead’s resource equation indicates that an average of at least six is correct.

They are 24 mice’s, 12 experimental and 12 controls. (Line 58).

However, in Figure 1, there are 21 pregnant mice. Not all mice got pregnant?. Please clarify this difference.

12 mice per group were used in the first study, while about 10 mice per group were used in the second study.  We now label the two studies more clearly as “study 1” and “study 2” in lines 61, 66, 97, and 110.  We also indicate in lines 74-76 that Figure 1 shows the scheme used to collect data on day 18 of pregnancy in study 2.  A similar approach was used to gather these data on day 15 in study 2.

It should be noted that the 95% Confidence Intervals have been calculated.

This is noted in lines 91-92.

Results

Experimental and control mice consumed food and gained weight at the same rates during pregnancy (data not shown). Please indicate these results.

These data are presented in lines 96-97.

This phrase:

"The effect sizes for ANOVA of these data on a per dam or per offspring basis were r = 0.77 and 0.58, respectively, which are of crucial practical importance [21]. "

It is not correct in results, since it is a subjective assessment, more appropriate to the discussion. Not applicable in results.

These sentences have been moved to the discussion (lines 180-184).

Table 1, please if you use three decimal places in all or if you use two decimal places, use the same criteria in all the results.

Done between lines 107 and 108.

At the bottom of the table, indicate which test you have used.

Shown in lines 108-109.

In Figure 2, 3 4, 5, 6 and 7 the mean plus minus 95% CI is not represented. Please check it out. Indicate at the foot of the figure the tests used.

We now show in the Y axis Figure labels “mean + 95% CI.”  Tests used are shown in the figure legends.

Discussion

Please adjust to the results obtained. They should review the knowledge on this topic with the literature. As well as indicating the strengths and limitations of this study. Incorporate guidance from future research based on the results obtained.

Also include how this study can help to know the effects in humans, given the species barriers.

We added a Limitations section.

Please also add separate conclusion section

We added a Conclusions section.

Round 2

Reviewer 2 Report

Thank you very much for allowing me to revisit the revised article and the authors' response "Consumption of a branched-chain amino acid (BCAA) during days 2-10 of pregnancy causes abnormal fetal and placental growth: implications for BCAA supplementation in humans" (ijerph-765661).

The authors have clarified many of the proposed points and I think that the understanding of their work has been improved.

Comments
The introduction has not been reinforced with the pathophysiological basis of the intake effect of BCAAs during the pregnancy and the efccect on outpring.
The hypothesis is prior to the objective, I suggest changing the position.
Material and methods: lack of sample size is a major weakness of the study, I suggest that you try to improve this point in future studies. Their reasoning for sample size is highly subjective.
If there are two studies, Study 1 with 24 mice and Study 2 with 20 mice, what does Figure 1 represent with 21 mice?
The figures still do not represent the 95% CI, or they change it or must add this information in the figures.
In the discussion, removing the adjectives "Crucial" ... is not scientific.

Author Response

Reviewer 2

Thank you very much for allowing me to revisit the revised article and the authors' response "Consumption of a branched-chain amino acid (BCAA) during days 2-10 of pregnancy causes abnormal fetal and placental growth: implications for BCAA supplementation in humans" (ijerph-765661).

The authors have clarified many of the proposed points and I think that the understanding of their work has been improved.

Comments

The introduction has not been reinforced with the pathophysiological basis of the intake effect of BCAAs during the pregnancy and the efccect on outpring.

As we state in lines 52-54, and to our knowledge, nothing more is currently known about how Ile might influence embryo development.  To our knowledge, the same is true for all three BCAAs.

The hypothesis is prior to the objective, I suggest changing the position.

We are sorry that we do not seem to understand this point.  In our view, testing our hypothesis was and is our objective.

Material and methods: lack of sample size is a major weakness of the study, I suggest that you try to improve this point in future studies. Their reasoning for sample size is highly subjective.

Historically about six animals have been used per treatment group in such studies [e.g., reference 12].

If there are two studies, Study 1 with 24 mice and Study 2 with 20 mice, what does Figure 1 represent with 21 mice?

As we previously stated in line 66, our intention was to have about 10 dams in each group in study 2.  In practice, not all groups of mice mated and began their protocol at precisely the same time for each mouse since mice were not necessarily on the same cycle when placed with males.  And some mice mated but did not become pregnant.  As a consequence, our group sizes ranged from 8 to 12 dams, as we now state explicitly in line 66.  We replaced the words “about 10” with the words “8 to 12” in line 66.

The figures still do not represent the 95% CI, or they change it or must add this information in the figures.

We apologize for still not understanding your point.  The 95% CIs were calculated for each set of our data, and those 95% CIs are reported in our figures as indicated on the Y axis of each figure.

In the discussion, removing the adjectives "Crucial" ... is not scientific.

“Crucial” is a descriptive term used in reference 21, which was written by experts in statistics.  In our view, it is a better word than the word “large,” which the authors also use in reference 21.  We can change the word “crucial” to the word “large” if it better fits the journal.

This manuscript is a resubmission of an earlier submission. The following is a list of the peer review reports and author responses from that submission.

Round 1

Reviewer 1 Report

ijerph-725858

Thank you very much for allowing me to review the article "Consumption of a Branched-Chain Amino Acid (BCAA) Alters Fetal Growth: Implications for BCAA Supplementation to Alleviate Muscle Damage and Soreness During Intense Human Exercise Training" (ijerph-725858).

I suggest a clearer title; it should be stated that it´s supplementation during pregnancy. The title really is confusing. Please clarify.

The phrase: "However, perturbations in amino acid and protein consumption have unwanted transgenerational effects on male and female reproduction", it is not based on evidence, therefore it should be considered as a hypothesis, that is ... could ...

This is an experimental study in mice during their pregnancy.

Please indicate the approval of the ethical committee of animal experimentation.

Introduction

I believe that a scientific journal should not use websites, which are not from official organizations. I suggest that you replace web pages with bibliographic citations from scientific journals, which give credibility to their contributions.

Both the summary and the end of the introduction clearly indicate the objective of the work. Please clarify it.

The methods section should be better explained, especially the collected variables and the design carried out.

A flow chart indicating the mice that participate in each group would be clarifying.

Results
Please indicate the sample size the results are not clear, these are confusing. (12 vs 12).

The discussion is very scarce.

There is an important aspect that confuses and is that one reason has many mouse fetuses and the unit of analysis, pregnancy or fetus is confused?

Reviewer 2 Report

The authors studied the effect of BCA on placental and fetal weights. Large speculations in the discussion and needs to be supported by the works done in the nutrient transport. The authors must consider including the following papers in the discussion.

   Endocrinology. 2011 Mar; 152(3): 1119–1129. Published online 2011 Feb 1. doi: 10.1210/en.2010-1153

J Physiol. 2013 Feb 1; 591(Pt 3): 609–625. Published online 2012 Nov 19. doi: 10.1113/jphysiol.2012.238014

The main effect of leucine supplementation is to activate the mTOR signaling. Numerous studies suggest that mTOR signaling function as a nutrient sensor in the placenta and per se regulates the fetal growth. The authors need to mention the previous works to discuss the effect of Ileu on fetal growth.  

Amino acid transport is another factor that regulates fetal growth and author should consider to emphasis the amino acid transport effect on fetal growth.      

Does any sex effect on fetus and placental weight? if so what is that and if not briefly mention the sex programming effect on fetal growth.    

Please avoid mentioning the E mice term and refer the term by their experimental groups.  

Include the distribution curve for the fetal weight instead of the bar chart.   

The rationale for choosing the Ileu is not clear and need to mention in the introduction.    

The methods are not clear and are need to be elaborate.  General write up is not clear and please avoid jargon and clear in the concept.